# Immunostaining for VEGF and Decorin Predicts Poor Survival and Recurrence in Canine Soft Tissue Sarcoma

**DOI:** 10.3390/vetsci10040256

**Published:** 2023-03-28

**Authors:** Jonathan P. Bray, Matthew R. Perrott, John S. Munday

**Affiliations:** 1AURA Veterinary, 70 Priestley Road, Surrey Research Park, Guildford GU2 7AJ, UK; 2School of Veterinary Science, Massey University, Palmerston North 4410, New Zealand

**Keywords:** soft tissue sarcoma, prognosis, canine, angiogenesis, VEGF, decorin

## Abstract

**Simple Summary:**

Soft tissue sarcomas are one of the most common malignant tumours affecting the dog. Surgical removal is curative for the majority of patients, but local recurrence can develop in almost 20% of patients. Local recurrence is the most common reason for euthanasia, and a significantly shorter survival time. A recurrent tumour most likely arises from isolated tumour cells that remain in the wound bed following removal of all of the visible tumour. This study seeks to identify whether the presence of two proteins—vascular endothelial growth factor (VEGF) and decorin—within the tumour may assist with prediction of tumour recurrence. Both proteins are known to play important roles in enabling development of new blood supply to a developing tumour. It was hypothesised that if the original tumour contained proportions of these proteins known to be important for vascular development, this may provide residual cancer cells with a more enhanced ability to recrudesce into a visible recurrence. This study showed that recurrence following surgery was more likely if the tumour had a high level of staining for VEGF and/or did not stain for decorin. This study supports the hypothesis, but further investigation is required to validate this finding.

**Abstract:**

The aim of this study was to investigate whether using immunohistochemistry to detect the angiogenic proteins vascular endothelial growth factor (VEGF) and decorin can help predict the risk of local recurrence of, or death from, canine soft tissue sarcoma (STS). VEGF and decorin were detected using validated immunohistochemical methods on 100 formalin-fixed paraffin-embedded samples of canine STS. The tumours had been resected previously, with clinical outcome determined by questionnaire. Each slide was assessed by light microscopy and the pattern of immunostaining with VEGF and decorin determined. Patterns of immunostaining were then analysed to detect associations with outcome measures of local recurrence and tumour-related death. High VEGF immunostaining was significantly (*p* < 0.001) associated with both increased local recurrence and reduced survival time. The distribution of decorin immunostaining within the tumour was significantly associated with survival time (*p* = 0.04) and local tumour recurrence (*p* = 0.02). When VEGF and decorin scores were combined, STS with both high VEGF and low decorin immunostaining were more likely to recur or cause patient death (*p* < 0.001). The results of this study suggest that immunostaining of VEGF and decorin may help predict the risk of local recurrence of canine STS.

## 1. Introduction

Soft tissue sarcoma (STS) occur commonly in the dog, and have been reported to represent 8–15% of all cutaneous or subcutaneous tumours [1]. Almost one in five dogs with a STS will die from their tumour [2,3,4]. Local recurrence after surgery remains the most challenging aspect of tumour management, with recurrence associated with a significantly reduced survival for affected patients [4,5].

Local recurrence is reported to occur more commonly with high-grade STS or when resection margins are incomplete [3,4,5]. Incomplete margins are detected by the presence of neoplastic cells extending to the tissue margins. Interestingly, not all STS that have incomplete histological margins recur after excision [6]. Why some STS recur but not others is not fully understood. However, it is hypothesised that the ability of a cancer cell to stimulate angiogenesis may be a critical factor in tumour recurrence [7,8]. Angiogenesis is stimulated by a variety of factors such as vascular endothelial growth factor (VEGF), fibroblast growth factor (FGF), platelet-derived growth factor (PDGF), placental growth factor, and transforming growth factor-β (TGF-β). It follows that tumour recurrence may be dependent on the ability of any neoplastic cells remaining in the original tumour site to produce these factors [9]. As any residual tumour cell in the wound bed will be derived from the original STS, if an excised tumour contains high levels of these angiogenic factors, this suggests any residual neoplastic satellites will likewise produce high levels of these factors, making a tumour more likely to recur after excision. Conversely, few angiogenic factors in an excised tumour suggests any neoplastic satellites that may be present are less likely to be able to stimulate sufficient angiogenesis for recurrence of the excised STS.

For the present study, VEGF and decorin were selected as markers of angiogenesis potential for evaluation as prognostic markers. VEGF has been previously shown to promote tumour angiogenesis and increased VEGF within a neoplasm has been reported as a negative prognostic factor for a wide range of tumour types in humans [10,11]. VEGF has been previously studied in canine STS with VEGF immunostaining identified in about 65% of tumours [12]. In another canine study, the serum concentration of VEGF was shown to reduce following excision of the STS, suggesting the tumour was contributing to the increased VEGF production [13]. To the authors knowledge, VEGF has not been previously investigated as a prognostic marker for canine STS. In contrast, decorin is an inhibitor of angiogenesis [14,15]. In a study of human STS, lower decorin concentrations within a tumour were associated with a significantly shorter disease-free survival and overall survival times [16]. There have been no published studies on the influence of decorin and cancer in the dog.

The aim of the current study was to use immunohistochemistry to identify whether the presence of VEGF and decorin within a canine STS could be used to predict tumour recurrence or patient survival. It was hypothesised that increased immunostaining for VEGF within a STS would predict increased tumour recurrence rates and reduced patient survival times while high decorin would predict a more favourable prognosis.

## 2. Materials and Method

### 2.1. Patient Selection

This immunohistochemical study was performed using cases selected from a tissue archive of formalin-fixed paraffin-embedded (FFPE) specimens, termed the parent population. Cases were excluded from selection if the grade of the tumour or status of local recurrence was unknown. Because the archived population was known to have a high number of grade I tumours, stratified sampling using proportional allocation was used to select the cases used for immunohistochemical staining [17]. This was achieved by firstly determining the proportion of local recurrence that occurred for grade I, grade II and grade III STS within the parent population. This calculation was then used to decide the number of cases from each grade that should be selected from the parent population to create a total sample population of 100 dogs. The random function within R (R v 3.2.3, R Development Core Team) was used to select patients from each grade, based on these previously calculated proportions.

Clinical details about each STS had previously been determined by questionnaire [18]. Follow-up information available for each tumour included patient demographics such as breed, age, sex and neuter status; tumour size, location and palpable characteristics (fixed or mobile); and the current status of the dog including the period in days until the development of local recurrence, metastasis, or death. Resection margins performed by the surgeon were defined as marginal (resection performed immediately about the gross tumour boundary), local (incorporating a small, unmeasured section (0.5–2 cm) of normal tissue about the gross tumour), wide (measured 2–3 cm margin) or amputation. If this information was not available in the clinical records, the resection margin was defined as unknown. The histological diagnosis and grading characteristics, including information on the degree of differentiation, percentage necrosis and mitotic count of each STS, had been previously reviewed by a single pathologist according to published guidelines [5]. The mitotic count was defined as the number of mitotic figures in 10 contiguous high power fields (hpf) (400×) from the most cellular part of the tumour. Occasionally, more than one area of a tumor with variable cellularity was scanned when selecting contiguous areas was not possible [3]. Information on the status of the histological margin was not available for all patients, so this factor was not included in the analysis.

### 2.2. Immunohistochemistry

Tissue sections (5 μm) were obtained from each tumour and mounted onto positively charged glass slides. Sections were dewaxed in xylene and rehydrated in a graded alcohol series and equilibrated in phosphate buffered saline. Antigen retrieval was performed in a decloaker (Biocare Medical, Pacheco, CA, USA) at 100 °C for either 20 min (VEGF) or 2 min (decorin) in a citrate buffer solution (EnVision™ FLEX Target Retrieval Solution (high pH), Dako Australia Pty. Ltd., Sydney, Australia). Immunohistochemistry was then performed using a Sequenza Immunostaining Center (Thermo Fisher Scientific, Basingstoke, UK). Endogenous peroxidase activity was blocked using Peroxidase-Blocking Reagent (EnVision™ FLEX, Dako Australia Pty. Ltd.) for 15 min. Tissue sections were incubated overnight with a 1:300 dilution of mouse antihuman VEGF polyclonal antibody [0.33 µg/mL] (VEGF (A-20) sc-152: Santa Cruz Biotechnology Inc., Dallas, TX, USA) or a 1:400 dilution of mouse antihuman decorin polyclonal antibody [0.25 µg/mL] (Anti-DCN, Sigma-Aldrich Co, St Louis, MI, USA). The specificity of these antibodies for the canine proteins has been previously reported so validation of these antibodies was not required [19,20,21,22]. Antibody detection was performed using diaminobenzidine (DAB; Dako Australia Pty). Positive and negative controls were used for each batch of slides. Positive control tissues for VEGF were FFPE sections of canine haemangiosarcoma; for decorin, sections of skeletal muscle were used. For negative control tissues, the primary antibody was omitted.

### 2.3. Evaluation of Immunostaining

Each slide was assessed by light microscopy and immunostaining of either VEGF or decorin was determined. Immunostaining was only evaluated in areas of well-preserved tissue morphology and away from areas of necrosis, tissue edges and other artifacts. Two investigators (JB and KM) reviewed all slides independently and were blinded to other features of the tumour. Where disagreement was present, consensus was achieved by joint review.

Immunostaining using anti-VEGF antibodies was scored using a previously reported method [12]. Briefly, tumours were scored based on the proportion of cells showing evidence of VEGF immunostaining across 5 non-adjacent and non-overlapping 40× fields. An initial low-power (10×) survey of the entire slide was performed initially to ensure areas of necrosis, processing artefact or poor staining were avoided. A tumour was classified as having “low VEGF” if less than 75% of cells were immunostained, whereas a tumour in which more than 75% of cells showed immunostaining was classified as having “high VEGF” (Figure 1). Where distribution was not homogenous across the tumour fields, the highest score observed was assigned.

The presence of decorin was determined by evaluating the distribution of immunostaining within the tumour. A “type 1” pattern was assigned when decorin immunostaining was confined to the peritumoural margins. A “type 3” pattern indicated that decorin-labelled stroma saturated the entire tumour and intertwined closely about individual cells while a “type 2” pattern was applied to cases where isolated islands of immunostained stromal tissue penetrated the tumour at various locations (Figure 2). When a STS showed little to no immunostaining within the tumour, the presence of intense immunostaining in the peri-tumoural tissues provided a good positive internal control.

### 2.4. Statistical Evaluation

All statistical analyses were performed with software (RStudio 2022.07.2 + 576, Posit, Boston, MA 02210, USA). Local recurrence and death due to the tumour were defined endpoints of this study. Survival time was defined as a dog dying or being euthanased due to either local recurrence or metastasis. The disease-free interval was defined as the number of days from surgery until local recurrence was identified by the veterinarian. Any cases with an unknown finding within the category being analysed were not included in the statistical evaluation of that characteristic.

Fisher’s Exact Test analysis was performed to evaluate variations in immunostaining according to age, sex, tumour size, location, palpable characteristics, surgical excision margins, tumour grade, presence of necrosis and mitotic count.

The Kaplan–Meier method was used to compare survival times to assess the significance of association between the immunostaining characteristics of a tumour with VEGF and decorin with the outcome measures of local recurrence. Univariate Cox proportional hazard analysis was used to assess the association between immunostaining characteristics and other clinically relevant variables described above against both the disease-free interval. Hazard ratios (HR), 95% confidence intervals (CI) and their corresponding *p*-values were calculated. A value of *p* ≤ 0.05 was considered significant. Prognostic factors that on univariate analysis had a *p* value < 0.1 were further used to evaluate their independence by use of the Cox’s proportional hazards model. Backward selection methods were used to create a fixed-effects model, retaining only those values that had a *p* value of < 0.05.

Tumours were subsequently divided into six groups based on the combined immunostaining using both antibodies. Groups were defined by combining the prognostic scores for each antibody from the most unfavourable to the most favourable, as follows: VEGF-high and decorin type 1; VEGF-high and decorin type 2; VEGF-high and decorin type 3; VEGF-low and decorin type 1; VEGF-low and decorin type 2; and VEGF-low and decorin type 3. Statistical analyses were repeated as above to determine differences between these six groups of immunostaining characteristics.

## 3. Results

### 3.1. Patient Selection and Demographics

From the original patient archive of 350 patients, 17 were excluded as local recurrence was unknown. A further two patients were excluded as tumour grade was undetermined. This left a population of 331 dogs. Within this remaining “parent population”, local recurrence occurred in 72 (22%) patients, comprising 68 (58% of all recurrences) in grade 1, 26 (33%) in grade 2 and 6 (8%) in grade 3. To create a sample of 100 patients for immunohistochemical study, the parent population was stratified by these proportions. Therefore, 22 (22%) patients were randomly selected from the parent population where local recurrence was recorded, comprising 12 (55% of all recurrences) grade 1, 7 (33%) grade 2 and 3 (13%) grade 3 STS. A further 78 (78%) patients were then randomly selected from the parent population where local recurrence did not occur, comprising 45 (58% of all recurrences) grade 1, 26 (33%) grade 2 and 7 (9%) grade 3 STS (Table 1).

Follow-up times for the study cohort ranged from 117 to 2114 days, with a median follow-up of 808 days. Over 85% of the study population had a follow-up time of over 12 months, with 19 (23%) cases being followed for more than 3.5 years (1290 days). The overall mean survival time was 1489 days, with a median survival time of 1685 days (95% CI 1490–1879). Death was attributed to the STS in 22 cases (22%). In this selected cohort, tumour grade, size, palpable characteristics, or resection margins did not predict survival or tumour recurrence, likely due to stratification. Local tumour recurrence developed in 22 (22%) cases. The median time to recurrence was 370 days (range 57–1818 days). Local recurrence of the tumour had a highly significant influence on overall survival (median survival time 939 days vs. not reached; *p* > 0.001).

### 3.2. Immunostaining—VEGF

Immunostaining for VEGF was interpretable in 82 sections of STS with artefactual defects or a lack of positive internal controls preventing interpretation in 18 sections. Details of the tumours included in this group are outlined in Table 2.

Of the 82 tumours where immunostaining could be interpreted, 43 (52%) tumours were graded as low VEGF and 39 (48%) were high VEGF. There was a statistically significant association between VEGF immunostaining and the surgical resection margin; 21 of 26 STS that had a marginal excision were classified as high VEGF (*p* < 0.0001). All 5 of the STS that had more than 50% necrosis had high immunostaining for VEGF. This compares with 12 of 23 (52%) and 22 of 54 (42%) STSs that had up to 50% necrosis and no necrosis, respectively. There was no association between VEGF immunostaining and the following characteristics: the sex (*p* = 0.7) or neuter status of the dog (*p* = 0.1), tumour location (*p* = 0.7), tumour size (*p* = 0.3), palpable characteristics (*p* = 0.1), tumour histologic type (*p* = 0.4), grade (*p* = 0.5) or mitotic count (*p* = 0.2).

Low VEGF was significantly associated with a longer overall survival time (χ^2^ = 13.0, *p* = 0.0003; Figure 3). The median survival time for patients with a low VEGF could not be calculated as more than 50% of the dogs remained alive at the close of this study. The median survival time for dogs with STS that had high VEGF was 1294 days (95% CI 774–1813 days). Overall, 85% of dogs with STS that demonstrated low VEGF remained alive more than 2 years after surgery, with 80% surviving 5 years or more. In contrast, only 50% of dogs with STS that had high VEGF 50% were alive after 2 years. Having a STS that was classified as high VEGF increased the risk of death due to the STS by a factor of more than four (HR 4.6 (95% CI 1.8–11.5, *p* = 0.001). On multivariate analysis, only high VEGF (HR 8.6, *p* < 0.0001, 95% CI = 2.8–26.4) was found to be associated with survival.

A STS with high VEGF was also significantly more likely to recur after surgery than a STS with low VEGF (56% vs. 9%, *p* < 0.001; Figure 4). High VEGF STS were 7.3-fold (95% CI 2.5–21.4, *p* < 0.001) more likely to recur than tumours with low VEGF. More than 90% of patients with a low VEGF STS remained disease-free at 2, 3 and 5 years, compared to 51%, 25% and 3%, respectively, of dogs with high VEGF STS.

### 3.3. Immunostaining—Decorin

Decorin immunostaining could be evaluated in 83 STS sections. Assessment was not possible in 17 cases due to artefactual defects or a lack of positive internal controls. Demographic details of the tumours included in this group are outlined in Table 3.

Of the 83 STS, 27 (32%) had a type I pattern, 24 (29%) a type 2 pattern, and 32 (39%) a type 3 pattern of decorin immunostaining. Half of the 50 STS that were less than 5 cm in diameter had a type 3 decorin pattern and this pattern was significantly more frequent in these smaller tumours than in STS that were greater than 5 cm in diameter (3 of 21 (14%); *p* = 0.02). Additionally, there was a significant association between the decorin immunostaining pattern and the extent of necrosis within the STS. A total of 27 of 45 (60%) tumours that had low levels of necrosis displayed a type 3 pattern whereas in tumours that had >50% necrosis, all 5 displayed a type 1 pattern. The distribution of decorin immunostaining patterns was significantly different depending on the location of the STS (*p* = 0.03). For tumours of the trunk, 16 of 28 (57%) had a type 1 immunostaining pattern, compared to 1 of 5 (20%) and 11 of 51 (22%) of tumours of the head and limb, respectively.

A total of 12 of 27 (44%) dogs that had STS with a type 1 pattern of decorin immunostaining died due to their STS compared to 6 of 24 (25%) dogs and 4 of 32 (12.5%) dogs that had STS with type 2 or 3 pattern, respectively. Differences between the groups were statistically significant (𝜒^2^ = 7.7, *p* = 0.02; Figure 5).

Local recurrence occurred for 8 of 21 (38%) STS with a type 1 pattern, 7 of 21 (33%) STS with a type 2 pattern and 6 of 21 (29%) STS with a type 3 patterns of decorin immunostaining. Differences between the groups were not statistically significant (*p* = 0.5; Figure 6).

The decorin immunostaining pattern was significantly correlated with the histological grade of the STS with low-grade tumours more likely to have a type 3 pattern and high-grade tumours more frequently having a type 1 pattern (*p* < 0.001; Figure 7).

### 3.4. Analysis—Combined VEGF and Decorin

There were 71 cases for which both VEGF and decorin immunostaining could be interpreted. Decorin and VEGF immunostaining were not correlated (*p* = 0.9). When the favourable and unfavourable extremes of the combined scores were compared, a STS with both a high VEGF and type 1 decorin distribution had a significantly lower MST than a dog with a STS with a favourable prognostic combination (low VEGF and type 3 decorin distribution; 1086 days vs. NA, *p* < 0.001; Table 4). No tumour-related deaths occurred in 18 dogs with a low VEGF/type 3 decorin combination, compared to 6/12 (50%) deaths in STS with a high VEGF/type 1 decorin combination (log rank 16.7, *p* = 0.005). Similarly, only 1/17 (6%) STS with a low VEGF/type 3 decorin combination recurred, compared to 6/12 (50%) with a high VEGF/type 1 combination (log rank 22.3, *p* < 0.001; Figure 8).

## 4. Discussion

High VEGF immunostaining within a STS was significantly associated with tumour recurrence. Dogs with STS with high VEGF were 4-fold more likely to die due to the STS than dogs with STS with low VEGF. This is the first study to demonstrate an association between VEGF immunostaining and prognosis in canine STS. Only one other study investigating VEGF immunostaining within canine STS has been previously reported, but the presence of VEGF was not investigated as a possible prognostic marker [12]. In that study, VEGF immunostaining of intra-tumoural tissue was observed in 46.15% of STS; this was similar to the proportion of STS classified with a high VEGF score in the current study. In human STS, two studies have reported a positive correlation between increased VEGF expression and higher tumour grade, but were unable to confirm an association with clinical outcome due to insufficient data [23,24]. In the current study, no correlation between STS grade and VEGF score was evident. This may be due to the small number of high-grade tumours in this study cohort.

The present study also showed significant differences in the risk of death due to a STS between dogs with STS that had different amounts of decorin immunostaining. However, no statistical association was found between decorin pattern and local recurrence, possibly due to the small patient numbers in this study. Decorin has not previously been investigated in canine STS. There are also limited prognostic studies evaluating the role of decorin in humans with STS, although the ability for decorin to influence the behavior of human cancer has been reported in several in vitro and in vivo studies [14,25]. In one study of human STS, decorin was assessed in 85 different tumours by real-time quantitative PCR and immunohistochemistry [16]. In that study, low decorin expression within a tumour was associated with significantly reduced disease-free and overall survival rates. In addition, decorin expression in recurrent or metastatic STS was lower than in the primary lesions, supporting a hypothesis that these secondary tumours have a more aggressive phenotype than the original primary tumour [16].

The prognostic potential of combining the immunostaining results of both VEGF and decorin was also evaluated in the present study. When VEGF and decorin immunostaining classifications were combined, the ability to identify subsets of tumours with very favourable or very unfavourable outcomes was improved. Thus, a STS with a combination of poor prognostic scores was more likely to recur or cause death of the dog compared to a STS with the most favourable combination of prognostic scores.

There are two possible hypotheses for how variation in VEGF and decorin in the tumour could influence the risk for tumour recurrence. Firstly, a significant association between tumour necrosis and increased VEGF was observed in the current study. It has previously been reported that tumour necrosis can be caused by hypoxia within a neoplasm, which in turn has been shown to stimulate the production of VEGF [26,27]. Hypoxia has also been shown to stimulate VEGF expression. Across the entire tumour mass, there will be heterogeneity, with some areas attaining adequate vascularisation, and other areas where oxygen and nutrient delivery remains poor. In a hypoxic environment, generation of hypoxia inducible factor (HIF-1a) within the affected cell will drive metabolism towards anaerobic glycolysis, with increased production of lactic acid [28,29]. It has been shown that cells derived from a persistently acidic environment are often in a dormant state, with cell proliferation held in G0 phase by reduced CDK-1 activity. Such dormant cells are known to be more immune to destruction by chemotherapy and radiation therapy. Thus, while a persistently hypoxic tumour microenvironment will undoubtedly lead to a high percentage of cellular death and tumour necrosis in sections of the tumour, it will also favour a genotype that is more resistant to immune destruction and more capable of surviving in a dormant state. It is possible that the high VEGF detected in STS in the current study is simply a surrogate indicator for a tumour that has been more hypoxic during its development. The increased rate of recurrence in tumours with high VEGF may be because these STS harbour a higher concentration of dormant cells within the peri-tumoural environment. These dormant cells may survive within the tumour bed following surgical resection, becoming reactivated to grow once environmental conditions improve following resection and tissue healing [30,31]. As seen with VEGF, decorin immunostaining was also associated with tumour necrosis. As decorin reduces angiogenesis, it appears possible cell hypoxia inhibits decorin production to allow greater blood vessel proliferation. Interestingly, larger STS had lower decorin immunostaining. As larger tumours are more likely to develop areas of hypoxia, this could also be an effect of a hypoxia with a neoplasm.

A second explanation is that VEGF and decorin may also alter tumour behavior due to their influence on the tumour microenvironment. Studies in human neoplasms have revealed that both VEGF pathway components and decorin can change the microenvironment in a way that increases the development of neoplastic cell satellites in the surrounding tissue [32,33]. This suggests that the increased rates of recurrence and reduced survival observed in STS with high VEGF and low decorin levels in the current study could be due to a wider distribution of satellite cells radiating beyond the immediate circumference of the tumour. If there is an increased proportion of microscopic satellite lesions persisting in the wound bed after surgery, this could result in an increased potential for recurrence after surgery.

There are several limitations to the present study. One important limitation is that the immunostaining was only interpreted by two people. For a prognostic scheme to be successful in the general population, the assessment criteria for a tumour has to be well-enough defined to minimise inter-pathologist variability. While the immunostaining in the current study appeared easy to assess, additional studies using a larger number of pathologists in different settings is required to ensure consistency in interpretation. Additionally, as this was a retrospective study, the quality of follow-up information obtained for all patients is uncertain. Evidence for tumour recurrence was dependent on the owners returning to their veterinarian; the cause of death was also open to interpretation as almost all dogs were euthanased, with no postmortem. In addition, confirmation of tumour recurrence or cause of death was limited to an assessment by the veterinarian; no histologic confirmation of local recurrence or post-mortem examination was performed in any case. It is unfortunate that correlation of the immunohistochemical findings with the status of the histological margins was not possible in the current study due to the manner in which the study materials were archived. Incomplete histological margins are known to be associated with significantly shorter disease-free intervals [4,5] so the absence of this information may introduce bias to the conclusions. As a consequence, validation of this study using multiple pathologists assessing a wider population of STS would be of value. Finally, the number of cases examined in this study was small and these results may not be a generalisable to a larger population. The small number of patients also led to some erroneous findings in subgroup analysis. As an example, this study found that almost all STS resected by marginal excision had a high VEGF. This would appear to be a random finding, as there is no valid explanation for why VEGF immunostaining should be influenced by the resection margin. There was no apparent statistical association found between resection margins and outcome of local recurrence or survival in this study. Nevertheless, the influence of this apparent selection bias on the overall results of this study is difficult to predict.

## 5. Conclusions

In conclusion, the results of this study suggest that evaluation of VEGF and decorin immunostaining within a STS may provide important prognostic information although addition validation of these results is essential.

## Figures and Tables

**Figure 1 vetsci-10-00256-f001:**
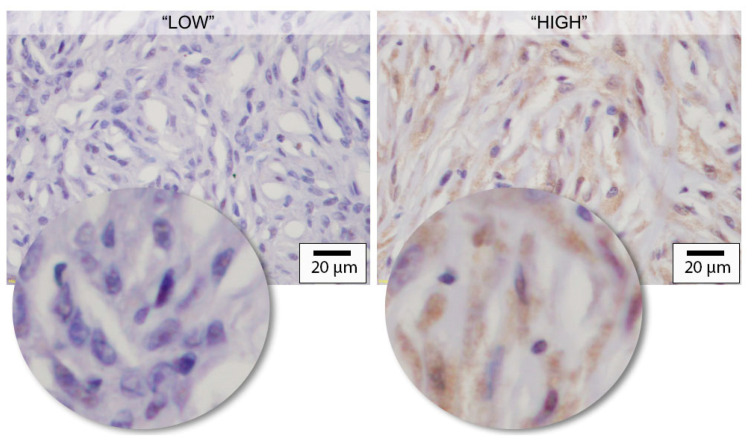
Grading scale of immunostaining for vascular endothelial growth factor (VEGF). The expected immunostaining pattern for VEGF was a strong, cytoplasmic stain. A low VEGF score was assigned if less than 75% of cells were immunostained. For a high VEGF tumour, more than 75% of the cells showed positive immunostaining (main image: 40× objective; inset images are unscaled digital enlargements for illustrative purposes).

**Figure 2 vetsci-10-00256-f002:**
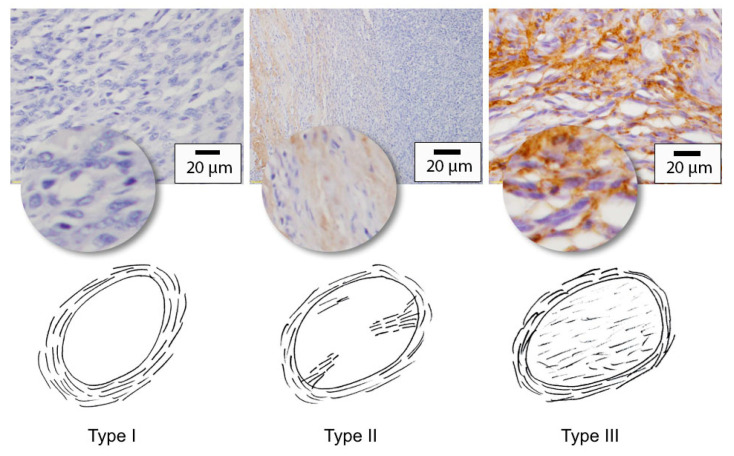
Grading scale for decorin immunostaining distribution pattern. Decorin immunostaining is typically observed within the stromal surrounding the cells. A type 1 pattern was assigned when decorin immunostaining was confined to the peri-tumoural capsule only and no staining was visible within the tumour itself. A type 2 pattern was applied to cases where isolated islands of immunostained stromal tissue penetrated the tumour at various locations. For a type 3 pattern, intensely staining decorin-labelled stroma saturated the entire tumour and intertwined closely about individual cells (main images: 40× objective; inset images are unscaled digital enlargements for illustrative purposes).

**Figure 3 vetsci-10-00256-f003:**
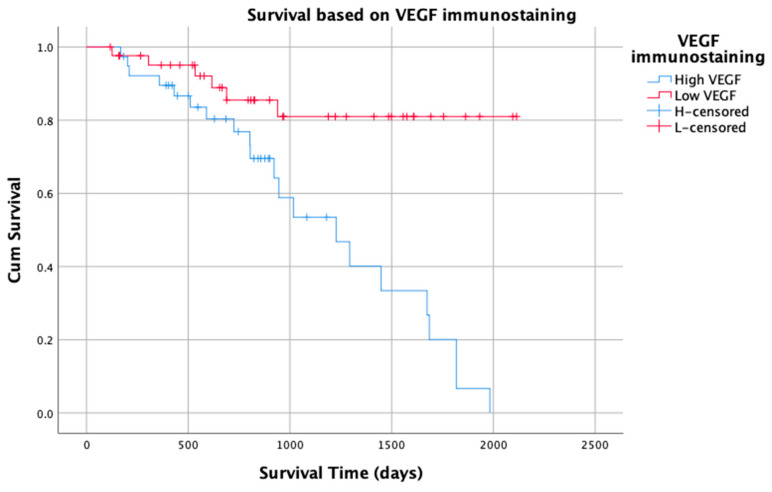
Kaplan–Meier graph of survival time for 82 patients with soft tissue sarcoma based on VEGF immunostaining (low and high).

**Figure 4 vetsci-10-00256-f004:**
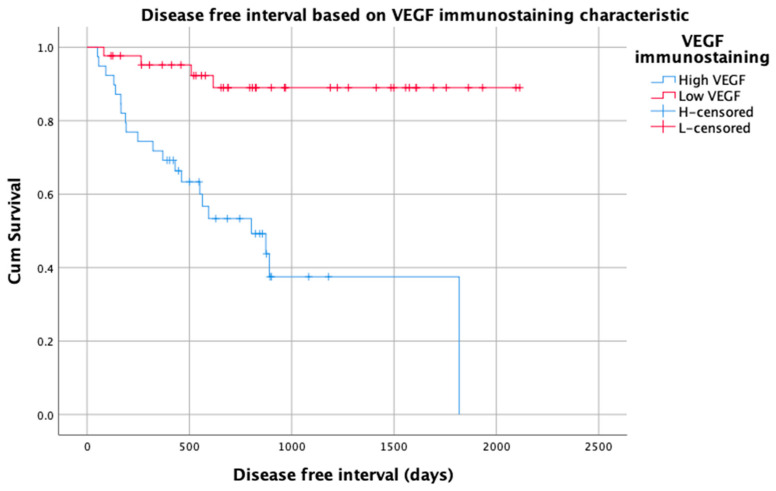
Kaplan–Meier graph of disease-free interval for 82 patients with soft tissue sarcoma based on VEGF immunostaining (low and high).

**Figure 5 vetsci-10-00256-f005:**
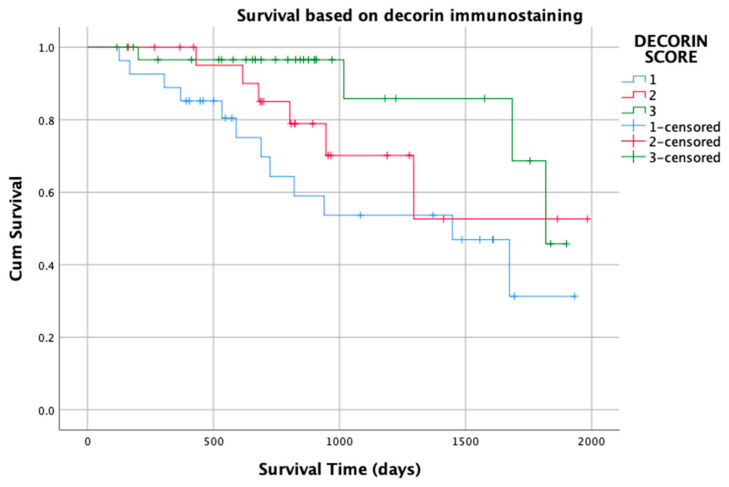
Kaplan–Meier graph of survival time for 83 patients with soft tissue sarcoma based on decorin immunostaining pattern (types 1, 2 and 3).

**Figure 6 vetsci-10-00256-f006:**
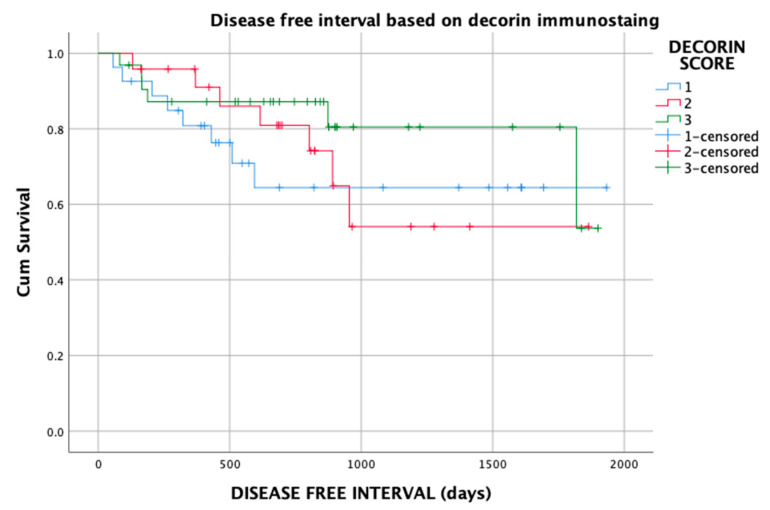
Kaplan–Meier graph of disease-free interval for 83 patients with soft tissue sarcoma based on decorin immunostaining pattern (types 1, 2 and 3).

**Figure 7 vetsci-10-00256-f007:**
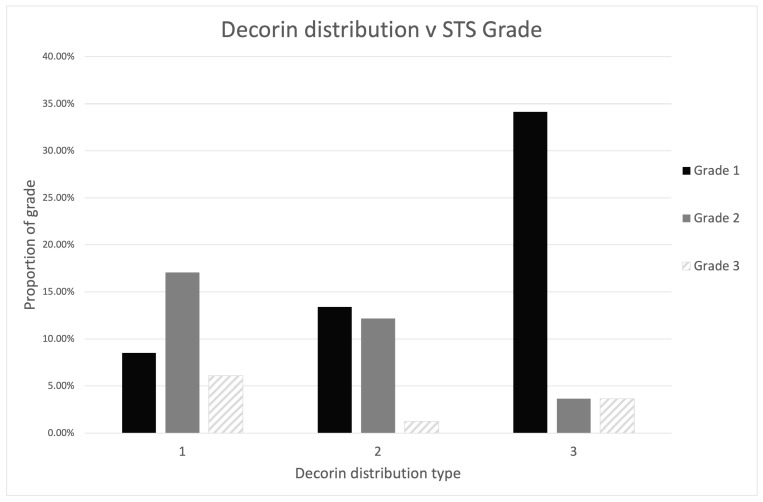
Distribution of decorin immunostaining pattern according to tumour grade. Low-grade tumours were more likely to have a type 3 decorin pattern.

**Figure 8 vetsci-10-00256-f008:**
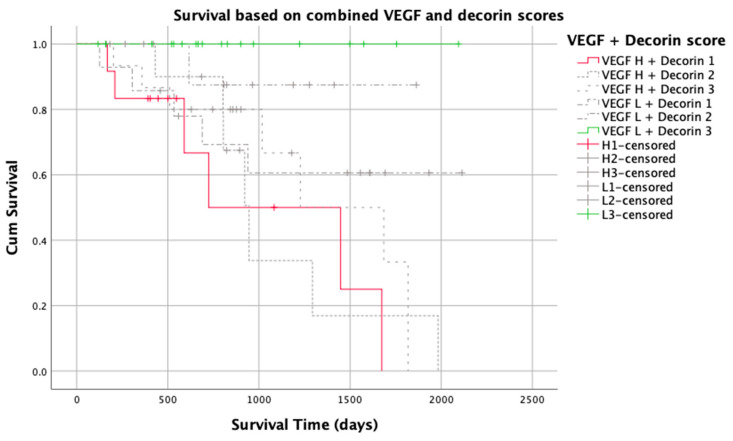
Kaplan–Meier graph of survival time for 73 patients with soft tissue sarcoma based on 6 groups of combined VEGF and decorin immunostaining levels. The extreme prognostic groups (H1 and L3) are highlighted; survival times between these two groups are significantly different (1031 days vs. 1924 days, *p* < 0.001).

**Table 1 vetsci-10-00256-t001:** Comparison of the rates of tumour grade and rates of local recurrence in the parent population and the study population after stratification. Patients were randomly selected from the parent population to provide equitable rates of recurrence by grade.

	Parent Population (331 Dogs)		Study Population (100 Dogs) (Stratified)	
	Local Recurrence	No Recurrence	Local Recurrence	No Recurrence
Grade 1	68 (58%)	176 (42%)	12 (55%)	45 (45%)
Grade 2	26 (33%)	67 (24%)	7 (33%)	26 (33%)
Grade 3	6 (8%)	16 (6%)	3 (13%)	7 (9%)
	72 (22%)	246 (74%)	22 (22%)	78 (76%)

**Table 2 vetsci-10-00256-t002:** Results from chi-square analysis for different tumour characteristics and VEGF immunostaining.

	VEGF (*n* = 82)			
		Low	High	
	*n* (%)	*n* = 43	*n* = 39	*p* Value
**Sex**				0.65
Female	49 (60%)	27	22	
Male	33 (40%)	16	17	
**Neutered**				0.13
No	45 (55%)	20	25	
Yes	37 (45%)	23	14	
**Tumour location**				0.75
Head	4 (5%)	2	2	
Trunk	27 (33%)	16	11	
Limb	51 (62%)	25	26	
**Size**				0.31
<1 cm	2 (2%)	1	1	
1–5 cm	45 (55%)	26	19	
>5 cm	21 (26%)	8	13	
unknown	14 (17%)	8	6	
**Palpable**				0.15
Discrete	32 (39%)	21	11	
Firmly adherent	45 (55%)	20	25	
Unknown	5 (6%)	2	3	
**Grade**				0.53
1	46 (56%)	25	21	
2	27 (33%)	12	15	
3	9 (11%)	6	3	
**Degree of resection**				0.0003
Marginal	26 (32%)	5	21	
Local	41 (50%)	30	11	
Wide	4 (5%)	3	1	
Amputation	7 (8%)	4	3	
Unknown	4 (5%)	1	3	
**Diagnosis**				0.11
Fibrosarcoma	20 (24%)	7	13	
Myxoma	1 (1%)	0	1	
Peripheral nerve sheath tumour	52 (63%)	29	23	
Perivascular wall tumour	9 (11%)	7	2	
**Tumour cause of death**				0.0001
No	55 (67%)	37	18	
Yes	27 (33%)	6	21	
**Local recurrence**				0.0001
No	60 (73%)	40	20	
Yes	22 (27%)	3	19	
**Survival Time**	days	days	days	
Minimum	117	117	168	
Maximum	2114	2114	1983	
Mean	907	961	847	

**Table 3 vetsci-10-00256-t003:** Results from chi-square analysis for different tumour characteristics and decorin immunostaining.

		Decorin (*n* = 83)			
		Type 1 (*n* = 27)	Type 2 (*n* = 24)	Type 3 (*n* = 32)	
	*n* (%)	*n*	*n*	*n*	*p* Value
**Sex**					0.02
Female	47 (57%)	21	13	13	
Male	36 (43%)	6	11	19	
**Neutered**					0.62
No	40 (48%)	11	12	17	
Yes	43 (52%)	16	12	15	
**Tumour location**					0.02
Head	5 (6%)	1	1	3	
Trunk	28 (33%)	16	3	9	
Limb	51 (61%)	11	20	20	
**Size**					0.03
<1 cm	0	0	0	0	
1–5 cm	50 (60%)	13	12	25	
>5 cm	21 (25%)	8	10	3	
unknown	12 (16%)	6	2	4	
**Palpable**					0.13
Discrete	31 (37%)	6	8	17	
Firmly adherent	45 (54%)	18	13	14	
Unknown	7 (8%)	3	3	1	
**Grade**					<0.0001
1	46 (55%)	6	11	28	
2	28 (34%)	13	12	3	
3	9 (11%)	8	1	1	
**Degree of resection**					0.14
Local	68 (82%)	21	20	27	
Wide	8 (10%)	3	4	1	
Amputation	7 (8%)	3	0	4	
**Diagnosis**					0.06
Fibrosarcoma	22 (26%)	11	2	9	
Myxoma	1 (1%)	0	0	1	
Peripheral nerve sheath tumour	52 (63%)	16	19	17	
Perivascular wall tumour	8 (10%)	0	3	5	
**Tumour cause of death**					0.02
No	61 (73%	15	18	28	
Yes	22 (27%)	12	6	4	
**Local recurrence**					0.55
No	62 (76%)	19	17	26	
Yes	21 (24%)	8	7	6	
**Survival time**		days	days	days	
Minimum	117	126	163	117	
Maximum	1983	1993	1983	1900	
Mean	884	928	877	882	

**Table 4 vetsci-10-00256-t004:** Univariate cox-regression analysis for survival and local recurrence for patients based on VEGF, decorin and combined immunostaining groups.

			Survival				Local Tumour Recurrence			
Immunostain (%)			Disease Free *n* (%)	Died from Tumour *n* (%)	*p*-Value	HR (95% CI)	No Recurrence *n* (%)	Local Recurrence *n* (%)	*p*-Value	HR (95% CI)
**VEGF** (*n* = 82)			55 (67%)	27 (33%)	>0.001	4.6 (1.9–11.5)			>0.001	7.3 (2.5–21.4)
Low 43 (52%)			37 (45%)	6 (7%)			39 (48%)	4 (5%)		
High 39 (48%)			18 (22%)	21 (26%)			17 (21%)	22 (27%)		
**Decorin** (*n* = 83)			61 (73%)	22 (27%)	0.041	0.5 (0.2–1.5)	62 (75%)	21 (25%)	0.214	
1: 27 (32%)			15 (18%)	12 (14%)			19 (23%)	8 (10%)		
2: 24 (29%)			18 (22%)	6 (7%)			17 (20%)	7 (8%)		
3: 32 (39%)			28 (34%)	4 (5%)			26 (31%)	6 (7%)		
**VEGF + Decorin** (*n* = 73)										
(Best)	L3	16	16 (100%)	0 (0%)	0.005		16 (100%)	0 (0%)	0.0005	
	L2	9	8 (89%)	1 (11%)			8 (89%)	1 (11%)		
	L1	11	8 (73%)	4 (36%)			9 (82%)	2 (18%)		
	H3	14	7 (50%)	7 (50%)			7 (50%)	7 (50%)		
	H2	11	4 (36%)	7 (64%)			4 (36%)	7 (64%)		
(Worst)	H1	10	5 (50%)	5 (50%)			5 (50%)	5 (50%)		

## Data Availability

The data presented in this study are available in Appendix A combined.

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
