# Peer review of "Immunostaining for VEGF and Decorin Predicts Poor Survival and Recurrence in Canine Soft Tissue Sarcoma"

_vetsci, 2023, doi:10.3390/vetsci10040256_

Round 1

Reviewer 1 Report

Dear Authors,

Interesting and very detailed work, on a very large number of cases. It provides valuable information on the possibility of recurrence and survival time in the case of soft tissue sarcomas, of which there are a lot and I believe that every year there are more and more.

I can see that the subject is not alien to the authors, because they are well-versed in literature.

An additional advantage of this work is the honesty of the authors who notice the weak points of the work. Unfortunately, it is sometimes difficult to get all the clinical information we need about our patients as opposed to human patients.

However, I believe that a few small corrections should be made.

The summary should be a little more elaborate referring to the results and discussions. In my opinion, this type of sentence that is in the manuscript is a bit too little.

I also recommend improving the quality of Fig. 1 and Fig. 2. Both figures, especially the higher magnification, are difficult to read. The BAR designation is also illegible. It is worth to additionally write in the description of the photo at what magnification these photos are taken.

Reviewer 2 Report

Thank you for inviting me to review this interesting study. The authors describe immunohistochemical staining and scoring of a cohort of canine STS with a range of histological grades as well as known clinical outcomes for two markers VEGF and decorin. The hypothesis is that these may help to predict risk of local recurrence and patient death, and be potential prognostic markers in the future. However, I cannot see where the authors have assessed whether these markers will aid prognostication over and above the current histological grading system, which is a flaw in this current manuscript. The authors also do not appear to address any potential relationship between margin size and rates of local recurrence. Both of these features (histological grade and margin size) could potentially be confounding factors. With some revisions, hopefully these can be addressed but as it stands this manuscript needs major revisions.

.

Introduction

Line 38: should “Soft tissue sarcoma” actually be pleural in this sentence?

Lines 41-42: might sound better as “associated with a significantly reduced survival time” or do the authors mean survival rate here? It is not clear as it stands.

Line 55: should this read “this suggests any residual neoplastic satellites like likewise”?

Line 61: typographical error – should read VEGF not VEFG

Line 69: punctuation error with references “,[14].[15]”

Line 70: “lower decorin concentrations within a tumour were associated” or “was associated” - ?

Line 70: significantly shorter disease-free….is there a word missing here? Disease-free interval, time, rate?

Materials and methods

Lines 97 – 98: I am assuming these margins are as assessed at the time of surgery, but could this be stated more clearly please (or if it is histological tumour free margins, this needs to be stated). Would it also be possible to give a measurement or range for the “local” category since this information has been provided for the “wide” category?

Line 101: mitotic count is preferred to mitotic rate. Please include more detail regarding field size / total area and magnification for the mitotic count.

Line 132: please define “field”. What objective, and what field size / area. How were the fields selected, was the whole slide assessed at lower power to find areas with the greatest amount of positive staining?

Figure 1. Please include in the figure legend the magnification of the main images and insets, the scale bar is too small to read. Please also include some description of the staining pattern you should expect to see – cytoplasmic, membranous, nuclear, diffuse, patchy, weak, moderate strong, etc.

Figure 2. I really like the line diagrams illustrating the three patterns. As for figure 1, please include in the figure legend the magnification of the main images and insets, the scale bar is too small to read. Please also include some description of the staining pattern you should expect to see – cytoplasmic, membranous, nuclear, diffuse, patchy, weak, moderate strong, etc.

Line 166: mitotic count is preferred to mitotic index.

Was any form of multivariate analysis performed? Only, some of these variables might be confounding? How do the authors know whether the significant associations seen with the IHC staining patterns were not related to the histological grade for example? Or factors within the grading scheme? For example, both VEGF and decorin appear to be associated with necrosis, but necrosis is also part of the histological grading system.  

Lines 176-182: how was the relative weighting of VEGF and decorin decided here, i.e. is a high VEGF score more significant than the decorin type pattern?

Results

Was there any association between the histological grade and VEGF staining patterns?

Was there any association between the histological grade and outcomes, either local recurrence or survival?

Was there any association between the margin classification and outcome, particularly local recurrence?

Line 201: the text refers to VEGF immunostaining and to table 2, but table 2 is decorin results

Lines 206-207: significant association between VEGF immunostaining and resection margins, but I cannot see this in the discussion section. Why do the authors think this is a finding, what does it mean?

Lines 208-209: All five of the STS that had more than 50% necrosis had high immunostaining for VEGF – but these are likely to be higher histological grade tumours also, so is this a confounding factor. Also, it is a very small number of cases.

It would have been particularly useful to know how the different grade tumours stained for VEGF (and decorin) and how that then related to the outcomes – so, for example, were there grade 2 tumours which stained differently for VEGF and then went on to behave differently in the patient? That would be very useful prognostic data.

Line 238: this section is about decorin, and refers to table 3, but table 3 is univariate cox-regression analysis for survival and local recurrence for patients based on VEGF, decorin and combined – I think the authors may mean table 2?

Table 3 – I cannot understand table 3. The columns do not appear to have headings, and the rows do not always align.

Figure 7 shows grade 1, 2 and 3 tumours for decorin patterns 1 and 3, but not pattern 2. Where is pattern 2 please? And why was grade and any association or absence of any association not mentioned for VEGF please?

Figure 8: I don’t think this graph needs to be included.

For the combined score, again there is no attempt to look at this in combination with other factors, in particular the histological grade and the margin size, which unfortunately detracts from the findings.

Discussion

Line 300: in the previous study only 65% of STS demonstrated VEGF staining. Is this comparable to the current study, and were those cases classified as “low”? Or did all of the STS in the current study demonstrate some positive staining, in which case, why the differences between the two studies?

Lines 302-303: positive correlation between increased VEGF expression and higher tumour grade – was there the same association in this study. Any findings in terms of grade and VEGF appear to have not been reported here. Was the same correlation seen in this study? If not, why the differences between the two studies?

Line 312: comma after in vivo studies, should this be a full stop?

Lines 324-326: I don’t feel there is enough evidence to support this here, as any additional prognostic markers would need to have a demonstrable benefit in terms of prognosis over and above the histological grade and the margin size. For some reason, this data has not been reported in this manuscript, despite it being available for the cases used in the study?

Line 327: decorin suddenly has a capital D

Necrosis appears associated with immunohistochemical staining both with VEGF and decorin. However, it is also a factor in the current grading system, so how does this help us with prognostication please?

One limiting factor with studies of this type is the absence of histological confirmation of local recurrence, metastatic disease and of post mortem examination – this needs to be mentioned in the discussion please.

Round 2

Reviewer 2 Report

Thank you for taking on board all of my previous comments. I just have one very minor comment left - regarding the size of area the MC was performed in. This is typically 2.37mm2 but it does vary between microscopes / digital platforms. Not wishing to appear pedantic, it is vitally important such information is included so that other researchers can accurately repeat your methods in the future. 

Author Response

Thank you for taking on board all of my previous comments. I just have one very minor comment left - regarding the size of area the MC was performed in. This is typically 2.37mm2 but it does vary between microscopes / digital platforms. Not wishing to appear pedantic, it is vitally important such information is included so that other researchers can accurately repeat your methods in the future. 

I understand. The challenge I have is that the original pathologic review of the sarcoma cases was performed by Keith McSporran in 2010-11. He has now retired, and is not contactable. I do not know what microscope set up he was using at that time. I have taken the methodology of MC assessment from his paper (McSporran KD. Histologic grade predicts recurrence for marginally excised canine subcutaneous soft tissue sarcomas. Vet Pathol. 2009 Sep;46(5):928-33. doi: 10.1354/vp.08-VP-0277-M-FL) as this would best reflected his routines at that time. However, I am unable to provide the precise area used.

If necessary, I can modify the text to say the precise objective area is not available?

Thanks

Jonathan